# Metabolic Reprogramming and Its Relationship to Survival in Hepatocellular Carcinoma

**DOI:** 10.3390/cells11071066

**Published:** 2022-03-22

**Authors:** Qingqing Wang, Yexiong Tan, Tianyi Jiang, Xiaolin Wang, Qi Li, Yanli Li, Liwei Dong, Xinyu Liu, Guowang Xu

**Affiliations:** 1CAS Key Laboratory of Separation Science for Analytical Chemistry, Dalian Institute of Chemical Physics, Chinese Academy of Sciences, Dalian 116023, China; wangqq15@dicp.ac.cn (Q.W.); wangxiaolin@dicp.ac.cn (X.W.); liqi1808@dicp.ac.cn (Q.L.); liyanli@dicp.ac.cn (Y.L.); xugw@dicp.ac.cn (G.X.); 2University of Chinese Academy of Sciences, Beijing 100049, China; 3International Cooperation Laboratory on Signal Transduction, Eastern Hepatobiliary Surgery Institute, The Second Military Medical University, Shanghai 200438, China; tanyexiong@smmu.edu.cn (Y.T.); 08300700057@fudan.edu.cn (T.J.)

**Keywords:** metabolomics, lipidomics, metabolic reprogramming, hepatocellular carcinoma, prognosis

## Abstract

Hepatocarcinogenesis is frequently accompanied by substantial metabolic reprogramming to maximize the growth and proliferation of cancer cells. In this study, we carried out a comprehensive study of metabolomics and lipidomics profiles combined with gene expression analysis to characterize the metabolic reprogramming in hepatocellular carcinoma (HCC). Compared with adjacent noncancerous liver tissue, the enhanced aerobic glycolysis and de novo lipogenesis (DNL) and the repressed urea cycle were underscored in HCC tissue. Furthermore, multiscale embedded correlation analysis was performed to construct differential correlation networks and reveal pathologically relevant molecule modules. The obtained hub nodes were further screened according to the maximum biochemical diversity and the least intraclass correlation. Finally, a panel of ornithine, FFA 18:1, PC O-32:1 and TG (18:1_17:1_18:2) was generated to achieve the prognostic risk stratification of HCC patients (*p* < 0.001 by log-rank test). Altogether, our findings suggest that the metabolic dysfunctions of HCC detected via metabolomics and lipidomics would contribute to a better understanding of clinical relevance of hepatic metabolic reprogramming and provide potential sources for the identification of therapeutic targets and the discovery of biomarkers.

## 1. Introduction

Liver cancer is the sixth most common cancer and the third leading cause of cancer-related death worldwide in 2020, with 905,677 new cases and 830,180 deaths [1]. Among them, hepatocellular carcinoma (HCC) accounts for about 75–85% of all primary liver malignancies. The past decade has brought considerable advancements in the discovery of biomarkers for the diagnosis, prognosis and prediction of treatment responses of HCC patients [2,3]. However, the underlying molecular mechanism of HCC onset and progression remains elusive, which seriously impedes the effective translation of scientific advances into clinical practice.

The liver is the central metabolic organ, orchestrating the interplay of multiple metabolic processes and maintaining metabolic homeostasis. Hepatocarcinogenesis is frequently accompanied by substantial metabolic rearrangements to meet the requirements of exponential growth and proliferation, which has been considered as an emerging hallmark of malignancy [4] and has aroused extensive interest. Evidence linking alterations of cancer metabolism to clinically relevant characteristics such as disease progression and therapeutic liabilities is often based on molecular profiling platforms, like the genomic and proteogenomic profiles of key signaling and metabolic pathways in HCC [5,6] that don’t directly reflect the functional consequences of measurable interactions. As the end-point of biological information flux, the metabolome has been widely accepted as a key link between the upstream genomics, transcriptomics and proteomics and phenotype to understand the molecular context of a biological system. Due to the advantages of high sensitivity, high throughput, and broad coverage, mass spectrometry-based metabolomics has been placed at the forefront of biomarker and mechanistic discoveries for pathophysiological processes. We previously reported the metabolic characteristics and their potential in clinical diagnosis based on 50 sets of matched liver tissues including HCC tissues and adjacent and distal noncancerous tissues by liquid chromatography–mass spectrometry (LC-MS)-based nontargeted metabolomics, but metabolite coverage was restricted owing to the limited instrumental detection and metabolite identification at that time [7]. A recent investigation from Fiehn’s lab [8] focused on the lipid remodeling in HCC and contrasted the findings in blood and matched malignant and nonmalignant liver tissues. Indeed, emerging insights have revealed that metabolomics is not only complementary to other upstream omics but also a direct regulator of biological processes and phenotypes by interacting with and actively modulating other omics [9,10]. Corroborating this, a variety of oncometabolites involved in reprogrammed cancer metabolism have been identified, such as 2-hydroxyglutarate [11,12], fumarate [13], succinate [14], lactate [15] and polyamines [16], which creates promising metabolic vulnerabilities for therapeutic interventions.

In this study, a comprehensive metabolomics and lipidomics profiling of 166 paired tumor and adjacent liver tissues together with gene expression analysis of the TCGA database was conducted to identify the reprogrammed metabolic activities and potential bioactive metabolites in HCC. The disrupted urea cycle and lipid metabolism were emphasized by the integrative differential expression and correlation analyses. Additionally, a molecule panel based on the metabolic network alterations was identified to stratify the prognostic risk of HCC patients. To sum up, our study suggested that the metabolic dysfunctions of HCC detected via metabolomics and lipidomics could advance our understanding of the clinical relevance of reprogrammed hepatic metabolism and provide potential sources for the identification of therapeutic targets and discovery of disease biomarkers.

## 2. Materials and Methods

### 2.1. Subjects 

A total of 166 patients who underwent hepatectomy at the Eastern Hepatobiliary Surgery Institute of the Second Military Medical University (Shanghai, China) between June 2013 and June 2014 were recruited in this study. Among them, 30 patients have no available follow-up information. Detailed clinical information for all subjects and 136 patients with follow-up information are listed in Table 1. Hepatocellular carcinoma tissue (HCT) and paired adjacent noncancerous tissue (ANT) from all patients were collected during the operation. The tissue samples were immediately placed into liquid nitrogen after surgical resection and stored at −80 °C until analysis. The overall survival was defined as the time interval from the date of surgical treatment to death or date of the last follow-up. This study was approved by the ethics committee of the Eastern Hepatobiliary Surgery Hospital of the Second Military Medical University and conformed to the standards of Declaration of Helsinki. Written informed consent was provided by all patients. 

### 2.2. Nontargeted Metabolomics and Lipidomics Analysis

The LC-MS-based nontargeted metabolomics and lipidomics profiling of tissue samples were performed as described previously [17,18]. Briefly, 20 mg of tissue was homogenized and extracted with the methanol/methyl tert-butyl ether (MTBE) /water system [19,20] containing internal standards. After centrifugation for two-phase formation, a 300 µL aliquot from the upper layer of the solution was drawn for lipidomics analysis, and a mixture of 150 µL aliquot from the lower layer and 200 µL from upper layer was drawn for metabolomics analysis. To monitor the robustness of batch analysis, QC samples were constructed from all tissue extracts to reflect an aggregated metabolite composition. All samples were lyophilized and stored at −80 °C prior to analysis.

For LC-MS based metabolomics analysis, the samples were reconstituted in 80 µL acetonitrile/water (*v*/*v*, 1:4) and a 5 µL aliquot of each sample was injected into a BEH C8 and a HSS T3 column (2.1 mm × 100 mm, 1.7 µm particle size) (Waters Corp, Milford, CT, USA) coupled with Q Exactive HF mass spectrometry (Thermo Fisher Scientific, Rockford, IL, USA) in positive and negative ion mode, respectively.

For LC-MS-based lipidomics analysis, the samples were resuspended in 30 µL chloroform/methanol (2:1, *v*/*v*) and diluted with 60 µL of isopropanol/acetonitrile/water (30/65/5, *v*/*v*/*v*) for negative ion mode. For positive ion mode, a further 2 times dilution was performed. The injection volume was 5 µL. A BEH C8 column (2.1 mm × 100 mm, 1.7 µm particle size) (Waters Corp, Milford, CT, USA) coupled with Q Exactive mass spectrometry (Thermo Fisher Scientific, Rockford, IL, USA) was operated in both positive and negative ion mode.

After the data acquisition, automated peak detection and integration were performed by Tracefinder software (Thermo Fisher Scientific, Waltham, MA, USA). All the raw peak areas were further normalized by their corresponding internal standards and tissue weight. The metabolite annotation was based on our in-house database [18] and online databases (HMDB and Metlin). The lipid identification was achieved according to our prior study [21], which was based on the retention time, extract m/z and MS2 fragments in combination with LipidSearch software (Thermo Fisher Scientific, Waltham, MA, USA).

### 2.3. Gene Expression Analysis

The TCGA Liver Cancer cohorts were obtained from The Cancer Genome Atlas (TCGA) database, and the gene expression data by RNAseq and corresponding phenotypic information were downloaded from UCSC Xena (http://xena.ucsc.edu accessed on 11 October 2021). In total, 373 tumor tissue samples and 50 solid tissue normal samples were included in this study. Statistical significance was performed using a Mann–Whitney U test with false discovery rate correction. 

### 2.4. Statistical Analysis

All statistical analyses and visualization were conducted using Microsoft Excel, GraphPad Prism 8.0 or R software version 4.1.0, unless otherwise noted. Clinical characteristics are presented as mean ± SD or n (%) as appropriate. The multivariate partial least square discriminant analysis (PLS-DA) with 200 times permutation tests was conducted using SIMCA-P 14.1 (Umetrics, Sweden). For univariate analysis, a Wilcoxon Signed-rank test for matched samples was performed using MATLAB (R2014a, MathWorks, Natick, MA, USA) software to evaluate the difference of identified ion features and lipid classes between ANT and HCT. The Benjamini-Hochberg method was employed to control the false discovery rate (FDR). Adjusted *p* values below 0.05 were considered as significant. Because lipid alteration between ANT and HCT can differ based on acyl chain length and the unsaturation degree, lipids were grouped and further analyzed based on the numbers of carbon atoms and double bonds. A Spearman correlation analysis was performed to determine correlations between identified features and clinical parameters.

Subsequently, a multiscale embedded correlation network analysis (using R package MEGENA) [22] was used to illustrate the differentially correlated molecular pairs in distinct conditions to decipher the dysregulated pathway in HCT compared to ANT. Differential correlation was calculated using R package DGCA [23]. Only molecular pairs with significantly differential correlation (*p* < 0.05) were included for the following network construction. The hub nodes identified by MEGENA were further sieved based on their chemical classes and correlation to build a metabolic panel. And then the patients were divided into three different risk groups using the optimal cut-off points according to the quantile distribution. Finally, the Kaplan-Meier survival curve for the cases with different risk was generated.

## 3. Results

From June 2013 to June 2014, a total of 166 pairs of tumor and adjacent nonmalignant tissues from HCC patients were collected in this study to map out the metabolic landscape and assess the association of metabolic phenotypes with the prognostic risk. Among the 136 patients with follow-up information, 47 patients (34.6%) died and 89 patients (65.4%) survived during the follow-up after operation. The median follow-up was 48.5 months (range, 2–67.6 months). The primary aim of this study is to reveal the metabolic reprogramming in malignant transformation and explore the prognostic value of related features for HCC patients.

### 3.1. Comprehensive Characterization of the Liver Tissue Metabolome and Lipidome

In this study, LC-MS-based nontargeted metabolic and lipidomics analyses were performed in both positive and negative ion modes to characterize the metabolite and lipid profiles of liver tissues. The precision of the overall analytical method was evaluated by the calculation of coefficient of variation (CV) in quality control (QC) samples as shown in Appendix A. Finally, a total of 625 ion features with CV less than 30% were retained and identified, and they consisted of 128 metabolites and 497 lipids. These identified ion features were further categorized into subclasses that reflect their chemical structure and function, as shown in Appendix A.

To achieve a comprehensive overview of possible metabolic alterations between HCT and ANT, supervised discrimination models were established based on partial least squares discriminant analysis (PLS-DA). As shown in the score plot of PLS-DA (Figure 1A), there is a clear separation between the two groups, with accumulative R2Y = 0.755 and Q2 = 0.71, indicating that the model possessed a good fit and predictive power. No overfitting was observed from the cross-validation by a 200 times permutation test. The intercepts of R2 and Q2 in the model were 0.14 and −0.166, proving the model is reliable and effective and that there are great differences existing between the two groups. 

### 3.2. Metabolic Disruptions in HCT Compared to ANT

To identify the specific differential features in tumor tissue compared to nonmalignant tissue from the same HCC patient, a Wilcoxon matched-pairs signed rank test was further conducted on all detected metabolites, lipids, and the biologically meaningful indexes including metabolite ratios, lipid ratios and the total amount of each lipid class. The univariate analysis revealed that a total of 105 metabolites (38 elevated and 67 decreased) and 408 lipids (179 elevated and 229 decreased) had significant changes (adjusted *p* < 0.05) in HCC tissue as illustrated by volcano plots (Figure 1B,C). To give a holistic view of metabolic reprogramming in HCT, the major dysregulated pathways based on differential metabolites and lipids were mapped in Figure 2.

#### 3.2.1. Metabolites

The main metabolic alterations based on metabolites was related to carbohydrate metabolism, urea cycle and acylcarnitine metabolism. Firstly, we observed upregulation of glucose-6-phosphate, Fructose-1,6-bisphosphate and phosphoenolpyruvate (PEP) in HCC tissue compared with adjacent noncancerous tissue. Meanwhile, the significantly elevated expression of hexokinase 2 (*HK2*), glucose phosphate isomerase (*GPI*), aldolase, fructose-bisphosphate A (*ALDOA*), glyceraldehyde-3-phosphate dehydrogenase (*GAPDH*), phosphoglycerate kinase 1 (*PGK1*) and enolase 1 (*ENO1*) were found in the gene expression analysis based on the 423 liver tissue samples in the TCGA database. These findings demonstrated a shift in carbohydrate metabolism from oxidative phosphorylation to aerobic glycolysis, which is a predominant characteristic of tumor cell known as the Warburg effect [24]. Concurrently, the expression elevation of gene glucose-6-phosphate dehydrogenase (*G6PD*) and the decline of ribulose-5- phosphate were also found in this study, suggesting the upregulated pentose phosphate pathway and nucleotide synthesis. Besides, lower glutamine levels were detected in HCT than ANT accompanied by the increase of the glutamate/glutamine ratio and the glutaminase 1 (*GLS1*) gene expression level, probably owing to a high metabolic dependency on glutamine for the cancer cell survival and proliferation [25,26]. In addition, accumulated glucosamine-6-phosphate and N-acetylglucosamine-6-phosphate (GlcNAc-6P) suggested the increased flux of glutamine and glucose to the hexosamine biosynthetic pathway.

Another crucial metabolic dysregulation is the urea cycle, by which excessive nitrogen is converted into urea in liver. The gene expression analysis showed that almost all essential genes encoding the urea cycle enzymes have a lower level in tumor tissue including argininosuccinate synthase 1 (*ASS1*), argininosuccinate lyase (*ASL*), arginase 1 (*ARG1*)*,* ornithine transcarbamylase (*OTC*)*,* arginine deiminase (*ADI*)*,* and carbamoyl-phosphate synthase 1 (*CPS1*), indicating the downregulated urea cycle in tumor than the surrounding liver. Consistent with our prior study [27], a striking elevation in arginine and a significant decline in ornithine were found in HCT rather than ANT, while no significant change was found in the level of citrulline. These recurrent metabolite alterations were observed in cohorts from different medical centers, showing the significant dysregulation of the urea cycle in HCC.

The acylcarnitine system, as a pivotal mediator in cancer metabolic plasticity, is involved in the bi-directional transport of acyl moieties between cytosol and mitochondria, thus acting a critical part in the tuning switch between the glucose and fatty acid metabolism [28]. In this study, the changes of acylcarnitines detected in our previous report [7] were further validated. The levels of virtually all detected acylcarnitines with medium and long chains (except C8:1, C10:2, C18:2, C18:2-OH) were significantly accumulated in tumor tissue compared with the adjacent noncancerous tissue, while the levels of most acylcarnitines with short chain reversely decreased, including C0, C3, C4-OH, C5, C5:1, and C6 (Appendix A). Consistent changes were also observed for the ratios of C2/C0 and C3/C0. The elevation of C2/C0 and the reduction of C3/C0 in HCT suggested the reverse alteration of β-oxidation for even- and odd- numbered fatty acids. Moreover, an increase in the ratio of (C16 + C18)/C0 indicated higher carnitine palmitoyltransferase 1 (CPT-1) activity in HCT than ANT, implying an improving entry of fatty acid into mitochondria for oxidation. Besides, the elevated ratio of (C16 + C18:1)/C2 reflecting the activity of carnitine palmitoyl transferase 2 (CPT-2) was also observed, but was not of statistical significance. Taken together, the acylcarnitine changes driven by HCC reflected an imbalance between the production and consumption of energy, although the effects of related genes and enzymes still require further confirmation.

#### 3.2.2. Lipids

The liver has a central role in the acquisition, storage and consumption of lipids. Growing evidence demonstrated the importance of lipid metabolic reprogramming in hepatocarcinogenesis and tumor adaptation to unfavorable conditions [29,30]. To have an overview of aberrant lipid metabolism in HCC, we firstly investigated the difference of the total amount of each lipid class between HCT and ANT (Figure 3A). The result showed that ether-linked phosphatidylcholine (PC-O), free fatty acid (FFA), cholesteryl ester (ChE) and triacylglycerol (TG) had no significant changes, while ether-linked lysophosphatidylcholine (LPC-O), ether-linked lysophosphatidylethanolamine (LPE-O) and dihexosylceramide (Hex2Cer) were significantly upregulated. In addition, all other differential lipid subclasses with statistical significance were found downregulated, suggesting a depletion of lipid constituents for the rapid growth of tumor cells. To gain insight into the associations of the lipid biochemical structure with the malignant transformation, the differences based on the lipid acyl chain and unsaturation degree were further compared for various lipid classes, as illustrated in Figure 3B. Overall, the most lipid individuals with longer acyl carbon chains and lower unsaturation degree showed increased content in HCT than in ANT except phosphatidylglycerol (PG). As the precursor of cardiolipin (CL, the signature phospho-lipid of mitochondria), almost all of PG were found with a lower level in malignant tissue. Considering that the significant reduction of tetralinoleoylcardiolipin (TLCL, the major CL in mitochondria) and total CL were also observed in tumor tissues, it is reasonable to deduce that CL remodeling is responsible for the progression of HCC.

Fatty acids (FA) are the major building blocks of complex lipid species, and therefore aberrant FA metabolism is a vital component of lipid reprograming in HCC. Strikingly, relative to saturated fatty acids (SFA), all detected monounsaturated fatty acids (MUFA) and the polyunsaturated fatty acids (PUFA) with long acyl chains had a higher level in HCT, implying a shift toward unsaturation in HCC, which has been associated with poor prognosis [31]. The specific alteration of FA individuals together with phosphatidylcholine (PC) and TG are shown in Appendix A. Interestingly, the lipogenic index (16:0/18:2), desaturation index (18:1/18:0, 16:1/16:0) and elongation index (18:0/16:0) were all found elevated in cancer tissue in this study, suggesting enhanced de novo lipogenesis (DNL), the higher activity of stearoyl-CoA desaturase 1 (SCD1), and the elongation of very long chain fatty acids protein 6 (ELOVL6). Other than the alteration of lipid contents and correlative indices, we also explore the gene expression difference associated with lipid metabolic pathways in HCC. Consistently, the higher expression level of indispensable genes for DNL, including ATP citrate lyase (*ACLY*), acetyl-CoA carboxylase alpha (*ACACA*), and fatty acid synthase (*FASN*) were observed in tumor tissue, further corroborating the pronounced DNL during hepatocarcinogenesis (Figure 2). Nevertheless, the insignificantly changed gene *SCD* and downregulated gene *ELOVL6* were also found in tumor tissue from the TCGA database, indicating that further investigations based on large representative cohorts and cell experiments are required to clear the underlying complex mechanism in the process of FA desaturation and elongation. Furthermore, SFA- and MUFA-containing phosphatidylcholine (PC) were elevated in tumor tissue, while most PUFA-containing PC decreased. The similar significant change trend was found for phosphatidylethanolamines (PE). Of note, the increase in MUFA-containing PC has been uncovered as a crucial event related to the proliferative switch of hepatocytes and hepatocellular carcinogenesis in murine and human models [32]. Additionally, compared with paired nonmalignant hepatic tissue, we also found significantly increased TG with the number of double bonds equal to or less than two in tumor tissue, but not for those with more than two double bonds. The same disturbed pattern was displayed by diacylglycerol (DG) except DG 34:2.

### 3.3. Multiscale Embedded Correlation Networks to Reveal Dysregulated Modules Associated with Prognosis

In addition to the above, we also performed a Spearman correlation matrix to assess the association of metabolic alteration with pathologically relevant clinical factors (Appendix A). We observed that metabolites and lipids are closely related to AFP, maximum tumor diameter, tumor-node-metastasis (TNM) stage and the Barcelona Clinic Liver Cancer (BCLC) stage, which are known as the prognostic indicators in HCC [33,34,35]. This result indicated that metabolic and lipidomics profiling have the potential to stratify the prognostic risk of HCC patients.

Distinct from the differential expression analysis, differential correlation analysis focuses on the changes of correlation between molecule pairs under different states rather than on individual molecules, thereby facilitating the identification of disrupted co-regulated networks in biological systems. Herein, in order to provide an additional insight on metabolic pathway dysregulation during hepatocarcinogenesis, we identified the differentially correlated molecule pairs via the package DGCA and then constructed networks to look for the pathologically relevant metabolic modules in HCT relative to ANT using MEGENA. Four notable modules (Figure 4) were identified from the global network based on all detected molecule pairs, with a significant change between neoplastic and non-neoplastic conditions. Module I comprises TG (18:1_17:1_18:2), TG (16:0_17:1_18:1) and TG 53:3 as hub nodes predominantly connected to other TG by red lines (+/+−), indicating that the correlation between the connected TG became weaker positive in HCT relative to ANT. Module II, with the FFA 18:1 as the central hub, mainly consists of FFA, DG, LPE and LPC connected with red lines, which are clustered by classes, implying their proximity in the metabolic pathway and that they may be coregulated. Although several overlapping segments exist between modules III and IV, they were revealed with different hub nodes (PC 40:2 and PC O-32:1 for module III, ornithine, PC 40:2 and PC (18:0_20:2) for module IV), possibly suggesting a subtle and stable link between the PC and ornithine metabolism. Of note, a reduced positive correlation in HCT was observed between ornithine and other amino acids in model IV, e.g., proline, tryptophan and methionine, illustrating the complex regulation of amino acid metabolism during HCC transition.

Subsequently, eight hub nodes identified from the four modules were further screened according to the maximum biochemical diversity and the least intraclass correlation [36] (Figure 5A). Finally, a panel consisting of ornithine, FFA 18:1, PC O-32:1 and TG (18:1_17:1_18:2) was generated and used to derive a risk index by the linear combination of the HCT/ANT concentration ratios weighted by their corresponding coefficients in Cox regression. A total of 136 patients with follow-up information were divided into low-risk, medium-risk and high-risk stages by trisection cut-off points. Then, the Kaplan-Meier curve (Figure 5B) showed clear separation among the three different risk groups and the patients with higher risk had significantly lower overall survival rate (*p* < 0.001 by log-rank test). This result indicated that the panel could serve as a beneficial tool for risk assessment and prognostic stratification of HCC patients and the dysregulation of their relevant pathways may represent potentially targetable metabolic vulnerabilities of HCC.

## 4. Discussion

Over the past decade, emerging evidence has demonstrated that metabolic reprogramming is a hallmark of malignancy [4]. However, the underlying mechanism of how reprogrammed metabolism sustains tumor growth and which reprogrammed activities are most related to therapeutic liabilities is still obscure, thereby requiring increased attention. In this study, we performed a comprehensive metabolite and lipid profiling of 166 paired liver tissues in combination with the gene expression analysis of the TCGA dataset. This integrative approach with an appreciable coverage and sample size allowed us to reveal the stable metabolic dysregulation in HCC and identify the metabolic characteristics associated with the prognostic risk of patients.

Reprogrammed metabolic activities contribute to the risk of HCC onset. In this study, we firstly systematically explored the altered metabolic pathways based on the differential individual molecules in combination with gene expression analysis. The enhanced aerobic glycolysis and DNL, aberrant acylcarnitine metabolism and repressed urea cycle were emphasized, which provide potential preventative and therapeutic opportunities to the patients. A recent study has shown that a glycolysis inhibitor, 2-Deoxy-D-glucose (2-DG), has the threptic potential to reverse tumorigenicity and sorafenib resistance mediated by protein arginine methyltransferase (PRMT6) deficiency in HCC [37]. Another agent, PEGylated arginine deiminase (ADI-PEG20) targeting urea cycle, which can convert arginine to citrulline, has shown its efficacy in vitro and in in vivo studies of HCC. However, the result of phase III clinical trials did not demonstrate significant improvements in prolonging overall survival of patients with advanced HCC [38], pointing towards the importance of further deciphering the underlying mechanism of the complex metabolic processes. 

Given that the undetected metabolic changes exist when relying solely on individual molecule levels, the differential correlation network was used to dig out the underlying pathways and key regulators of important biological processes. Specifically, DGCA/MEGENA was applied to query individual molecule pairs with differential correlations for network construction and to identify pathologically informative modules and hubs. Of note, in line with the result of differential expression analysis, ornithine was identified as a hub node, further emphasizing the essential role of urea cycle disorder in HCC. Corroborating this, mounting studies have discussed the relevance of urea cycle disorder for cancer diagnosis and therapy [39,40]. Furthermore, the metabolism of FFA, DG, TG and PC was emphasized in the co-regulation network. It is tempting to speculate that a subtle linkage might exist in the urea cycle and lipid metabolism. In fact, a precious observation has shown that long-chain fatty acids have a suppressive effect on the gene expression of urea cycle enzymes in primary rat hepatocytes [41]. Another investigation proposed the urea cycle enzyme arginosuccinate synthase (ASS) as a critical physiological regulator of hepatic AMP-activated protein kinase (AMPK), effectively coupling lipid oxidation to urea cycle activity [42]. These findings highlight the exquisite orchestration by an intricate network of metabolic and cell signaling events in the liver. On the other hand, they may provide actionable new insights into the exploitation of metabolic vulnerabilities.

In addition, a prognostic panel of ornithine, FFA 18:1, PC O-32:1 and TG (18:1_17:1_18:2), generated from the hub nodes of the identified modules, was preliminarily proved to have the capacity to achieve the risk stratification of HCC patients in this study. Among them, ornithine was known as a constituent of ammonia-lowering agents like L-ornithine L-aspartate (LOLA) and ornithine phenylacetate (OP) for the treatment of acute-on-chronic liver failure and hepatic encephalopathy (HE). A recent double-blind randomized controlled trial suggested that the addition of LOLA to the already established armamentarium of lactulose and rifaximin leads to early recovery from HE and better short-term survival rates, implying that LOLA might have indirect hepatoprotective effects apart from ammonia removal [43]. Moreover, unsaturated fatty acids in representation of oleic acid have been demonstrated to inhibit the expression of tumor suppressor phosphatase and tensin homolog (PTEN) in hepatocytes through a mammalian target of the rapamycin (mTOR)/nuclear factor kappa B (NF-κB)-dependent mechanism [44]. Further investigation indicated that the unsaturated fatty acids-driven down-regulation of PTEN plays a critical role in the hepatoma progression [45]. Meanwhile, it is found that TG could act as a protective role under the tumor-relevant hypoxic conditions by releasing the unsaturated fatty acids to prevent a toxic buildup of saturated lipids [46]. Additionally, monosaturated PC has a closely positive correlation with hepatic proliferation and carcinogenesis [32]. Collectively, our prognostic panel could be a promising predictor for the risk assessment of HCC patients, although a large, multicenter validation cohort is still needed in the future.

## 5. Conclusions

In conclusion, a comprehensive metabolic and lipidomics profiling combined with gene expression analysis was performed to identify the metabolic reprogramming in HCC. Our work underscored the dysregulation of the urea cycle and the lipid metabolism in HCC by the integration of differential expression and correlation analyses. In addition, a prognostic panel based on the metabolic network alterations was identified to assess the prognostic risk of HCC patients. Overall, the metabolic dysfunctions of HCC detected via metabolomics and lipidomics expand the understanding of clinical relevance of hepatic metabolic rearrangements, thereby offering some exciting opportunities for the intervention and treatment of HCC.

## Figures and Tables

**Figure 1 cells-11-01066-f001:**
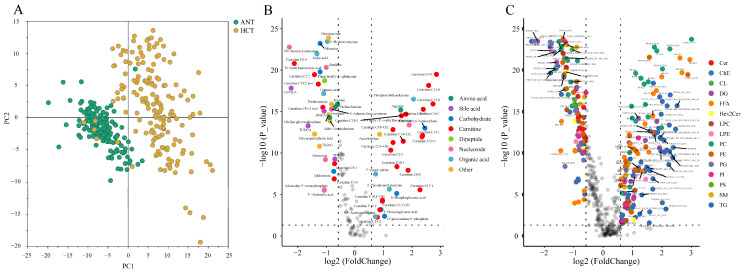
Comparative metabolomics and lipidomics profiles of adjacent noncancerous tissue (ANT) and hepatocellular carcinoma tissue (HCT) samples from HCC patients. (**A**) Score plot of PLS-DA model for ANT samples (orange dots) and HCT samples (green dots) separation (R2X = 0.177, R2Y = 0.755, Q2 = 0.71). Volcano plots of metabolomics (**B**) and lipidomics data (**C**) discriminating ANT and HCT samples. Log2 fold change values of normalized mean peak area are plotted against the respective −log10 transformed *p* values. Ion features with adjusted *p* < 0.05 were considered as significantly differential expression. The metabolites with |fold change| > 1.5 and lipids with |fold change| > 2 are highlighted with compound name. Detected compounds were compared using a Wilcoxon Signed-rank test for matched samples and the raw *p* values were adjusted to false discovery rate (FDR) using the Benjamini–Hochberg method.

**Figure 2 cells-11-01066-f002:**
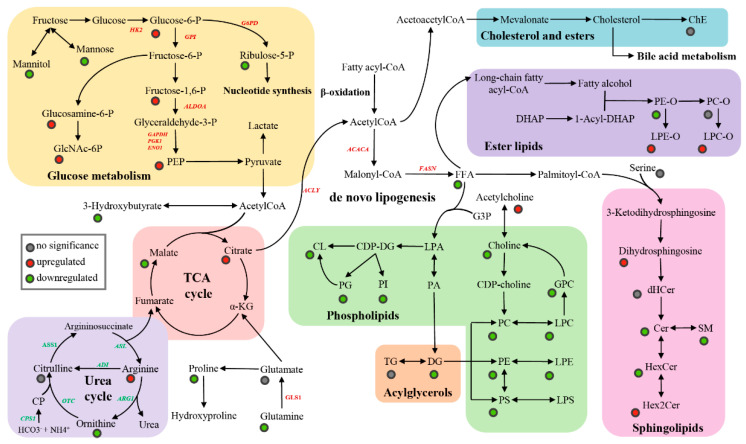
Overview of the lipid biosynthetic and metabolic pathways. Colored dots represent the metabolite and lipid changes in HCT samples compared to ANT samples. Note that the pathway network has been truncated due to space restrictions.

**Figure 3 cells-11-01066-f003:**
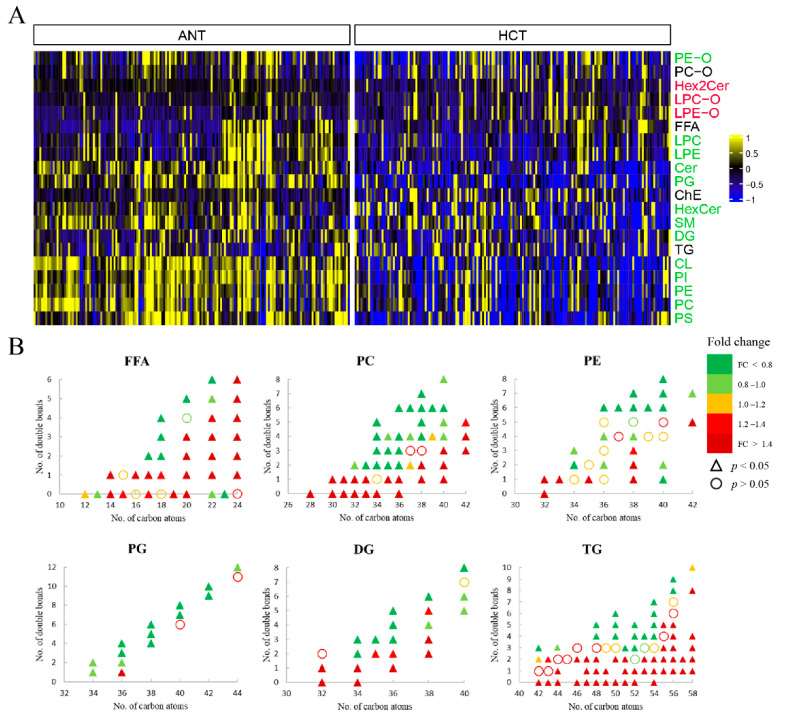
Association between HCC and lipid structure. (**A**) Heatmap of the lipid abundance changes by class between ANT and HCT samples. (**B**) Alteration of lipids with significant difference by carbon number and double bond number.

**Figure 4 cells-11-01066-f004:**
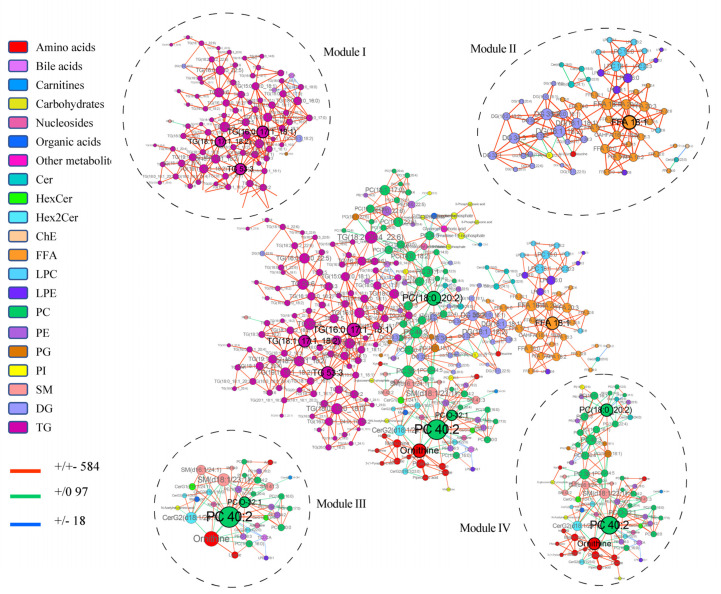
Differential correlation analyses of tissue metabolites and lipids in HCT relative to ANT. Only molecule pairs with significant differential correlations (*p* < 0.05) are included. Sign/sign indicates the direction and strength of the correlation in HCT/ANT, and the number that follows indicates the number of molecule pairs in the global networks exhibiting this pattern of change. For instance, the red line +/+− 584 indicates that correlation between two connected molecule pairs was positive (+) in ANT, and the correlation became weaker positive (+−) in HCT. A total of 584 molecule pairs connected by red lines in the global network displayed this pattern of change (+/+−). The hub nodes are labeled with black font.

**Figure 5 cells-11-01066-f005:**
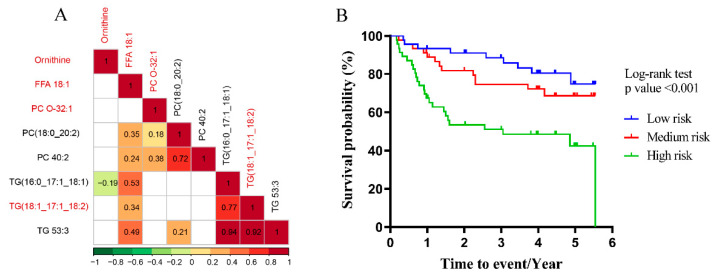
Identification of prognostic panel and the overall survival analysis. (**A**) The correlation matrix of hub nodes. Values and colors represent Pearson correlation coefficients of log-transformed metabolites with significance. The unsignificant correlations (*p* > 0.05) are filled with white. The molecules with red font were identified to generate a prognostic panel. (**B**) Kaplan-Meier curve of overall survival according to the panel of hub nodes identified by MEGENA.

**Table 1 cells-11-01066-t001:** Clinical and pathological features of HCC patients.

Characteristic		All (*n* = 166)	Patients with Follow-Up (*n* = 136)
Age, mean ± SD		49.3 ± 10.9	49.6 ± 11
Gender, n (%)			
	Female	21 (12.7%)	19 (14.0%)
	Male	144 (86.7%)	117 (86.0%)
	na	1 (0.6%)	0
Smoking, n (%)		80 (48.2%)	66 (48.5%)
Alcohol abuse, n (%)		33 (19.9%)	27 (19.9%)
Family History		34 (20.5%)	29 (21.3%)
HBsAg +, n (%)		136 (81.9%)	112 (82.4%)
AFP, >400 μg/L, n (%)		67 (40.4%)	55 (40.4%)
PLT (×10/L, mean ± SD)		164.5 ± 63.4	165 ± 63.3
TBA (umol/L, mean ± SD)		11.6 ± 15.4	10.6 ± 13.3
CEA (μg/L, mean ± SD)		2.8 ± 2.8	3 ± 2.9
CA19-9 (U/mL, mean ± SD)		19.6 ± 17.1	19.8 ± 17.9
Tumor Nodules, n (%)		69 (41.6%)	58 (42.6%)
MVI, n (%)		63 (38.0%)	54 (39.7%)
Multiple Tumor, n (%)		26 (15.7%)	19 (14.0%)
Maximum Tumor Diameter, mean ± SD		7.1 ± 4.6	7 ± 4.5
TNM Stage, n (%)			
	Ⅰ	76 (45.8%)	64 (47.1%)
	Ⅱ	39 (23.5%)	31 (22.8%)
	Ⅲ	11 (6.6%)	7 (5.1%)
	IV	39 (23.5%)	34 (25%)
	na	1 (0.6%)	0
BCLC Stage, n (%)			
	A	107 (64.5%)	91 (66.9%)
	B	19 (11.4%)	11 (8.1%)
	C	39 (23.5%)	34 (25.0%)
	na	1 (0.6%)	0
ALBI Grade, n (%)			
	1	122 (73.5%)	104 (76.5%)
	2	136 (81.9%)	29 (21.3%)
	na	8 (4.8%)	3 (2.2%)

Data are presented as mean ± SD or n (%) values as appropriate. na: not available. Abbreviations: HBsAg, hepatitis B surface antigen; AFP, alpha-fetoprotein; PLT: platelet; TBA: total bile acids; CEA, carcinoembryonic antigen; CA 19-9: carbohydrate antigen 19-9; MVI, microvascular invasion; TNM: tumor-node-metastasis; BCLC, Barcelona Clinic Liver Cancer; ALBI, albumin-bilirubin.

## Data Availability

The data presented in this study are available on reasonable request from the corresponding authors.

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
