# Peer review of "Metabolic Reprogramming and Its Relationship to Survival in Hepatocellular Carcinoma"

_cells, 2022, doi:10.3390/cells11071066_

Round 1

Reviewer 1 Report

The manuscript entitled "Metabolic reprogramming and its relationship to survival in hepatocellular carcinoma" describes a comprehensive study of metabolomics and lipidomics profiles combined with gene expression analysis to characterize the metabolic reprogramming in hepatocellular carcinoma (HCC).

In my opinion, the manuscript is well described and the results fundamented with the reported in literature.

For these reasons, the manuscript should be accepted after minor revisions.

Comments:

  • Abreviations should be described when used for the first time
  • Why LC-MS was chosen to analyze lipids?
  • What about recoveries of extraction method? And matrix effect?

Author Response

Response to Reviewer 1’s Comments

Point 1: Abreviations should be described when used for the first time.

Response 1: The description for all abreviations has been supplemented.

Point 2: Why LC-MS was chosen to analyze lipids?

Response 2: Lipidomics research is based on two main approaches currently: direct infusion mass spectrometry-based shotgun lipidomics and LC–MS-based lipidomics. Compared with shotgun approach, LC–MS enables highly efficient LC separation of lipids prior to MS detection, therefore reduces the complexity of the matrix. It typically provides a higher sensitivity and offers retention time as an additional parameter to identify lipid species. Due to the complexity of biological samples, LC-MS-based lipidomics was chosen to acquire lipid profiling in this study.

Point 3: What about recoveries of extraction method? And matrix effect?

Response 3: The extraction method we used in this study was developed and published in Journal of chromatography A, 2013, 1298: 9-16. Equal relative volume of upper and lower phases were combined for targeted and untargeted metabolomics analysis, and the metabolite recovery was comparable with classical 80% methanol-based metabolite extraction; an aliquot of upper phase was used for lipidomics analysis, and the lipid recovery was comparable with classical chloroform/methanol/H2O-based lipid extraction. It can be ovserved from Table 1 and Table 2 of the published paper (Journal of chromatography A, 2013, 1298: 9-16)  the recovery of (MTBE)/methanol/H2O-based biphasic extraction  is comparable with 80% methanol-based extraction. Matrix effect is common in biological metabolomics analysis, therefore, the signal intensity of analyte cann’t be directly compared between any two different types of matrices. While in the same matrix comparison, internal standards were used to correct the signal before comparison between different groups (healty vs. disease, or among HCC tissues, adjacent and distal noncancerous tissues) to neutralize the influence of some matrix effects.

Reviewer 2 Report

In general, the manuscript is well written and the reviewer found only some technical errors in the manuscript (see attached file).

All research activities were performed in detail.

Statistical processing of the obtained results and their categorization were also performed systematically.

The material and methods are very well and clearly written.

The results section is very nicely written with all the necessary and concise accompanying explanations.

All figures are well prepared, in a satisfactory resolution, so that all the stated and explained details are clearly visible.

The discussion part is excellent and explained to a completely satisfactory extent.

The references used are carefully selected.

Author Response

Response to Reviewer 2’s Comments

Point 1: In general, the manuscript is well written and the reviewer found only some technical errors in the manuscript (see attached file).

 Response 1: Thank you so much for your good words and help. All of them have been corrected.
